# Reliable Neighborhood-Aware Multi-View Outlier Detection

Huijie Ma [1]   Haoyuan Xin [1]   Lei Meng [2]   Guanzhou Ke [3]   Yongyong Chen [4]   Guoqing Chao [1]

## Abstract

In recent years, multi-view outlier detection (MVOD) has gained increasing attention, with the primary goal of recovering the underlying structure of normal data from outlier-contaminated multi-view datasets. However, this goal is hindered by two fundamental challenges:(1) outlier propagation, (2) scale discrepancy. To address these issues, we propose RNAMOD (Reliable Neighborhood-Aware Multi-View Outlier Detection), which introduces the concept of reliability and constructs a reliable neighborhood structure to avoid outlier propagation. We introduce a leave-one-out directional consensus mechanism to align cross-view neighborhood structures while preventing scale discrepancy by aligning geometric directions that remain invariant to scaling. Extensive experiments on six benchmark datasets demonstrate that RNAMOD consistently outperforms state-of-the-art methods.

## 1. Introduction

Multi-view learning (Brefeld et al., 2005; Chao et al., 2021; 2024) aims to use data information from multiple sources or perspectives to enhance data representation, thus enabling learned models to exhibit greater generalization capabilities. Multi-view learning has achieved significant success in multimedia applications (Jia et al., 2020; Yan et al., 2021).

Unsupervised outlier detection (Wyatt et al., 2022), also known as anomaly detection, aims to identify samples that deviate significantly from the majority of data in unlabeled datasets (Xie et al., 2016). It has achieved remarkable success in various applications, such as medical diagnosis (Wang et al., 2019) and spam email detection. However, most existing methods are designed for single-view data, limiting their applicability in complex multi-view scenarios where heterogeneous feature spaces exist. Moreover, these methods typically assume that outliers have little impact on global data structure learning, overlooking their negative influence during training. This contaminated learning may lead to *outlier propagation*, where abnormal samples distort neighborhood relationships (Zhang et al., 2026) and degrade detection performance.

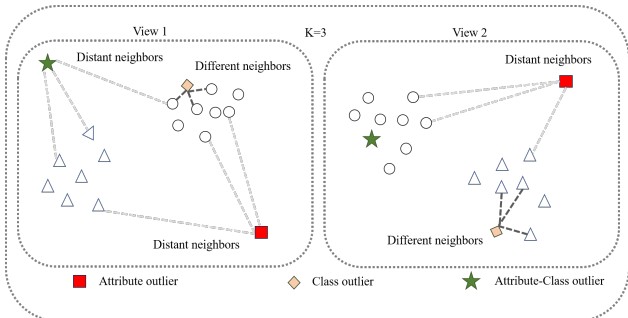

*Figure 1.* Description of Neighborhood-based outlier definitions: **Attribute outlier** (red) indicates instances that are remarkably distant from its neighbors within each view. **Class outlier** (orange) indicates instances with remarkably different neighborhood structures across views. **Attribute-Class outlier** (green) exhibits characteristics of both attribute outlier and class outlier.

In recent years, the emergence of multi-view outlier detection (MVOD) methods has effectively addressed the inherent limitations of single-view outlier detection, achieving notable success. Earlier MVOD studies are mainly clustering-based (Marcos Alvarez et al., 2013; Perozzi et al., 2014; Ji et al., 2019), which have drawn increasing attention due to their strong empirical performance. However, these methods rely on the assumption that the data exhibit clear clustering structures; when this assumption is violated, their effectiveness declines, and the definition of outliers becomes overly constrained by clustering priors. To relax this constraint, subsequent research has shifted toward neighborhood-based methods (Wang et al., 2023; Cheng et al., 2021), where the neighborhood structure is typically defined using the $k$-nearest neighbors ($k$NN). Unlike clustering-based models, neighborhood-based MVOD defines outliers without requiring explicit cluster separability, as illustrated in Fig-

---

[1]School of Computer Science and Technology, Harbin Institute of Technology, Weihai, Shandong, China [2]School of Software, Shandong University, Jinan, Shandong, China [3]Autel AI Lab, Autel Robotics, Shenzhen, Guangdong, China [4]School of Computer Science and Technology, Harbin Institute of Technology, Shenzhen, Guangdong, China. Correspondence to: Guoqing Chao <guoqingchao10@gmail.com>.

*Proceedings of the 43$^{rd}$ International Conference on Machine Learning*, Seoul, South Korea. PMLR 306, 2026. Copyright 2026 by the author(s).

ure 1. Accordingly, outliers in multi-view scenarios can be categorized into three types:

- **Attribute outlier**: Instances that are distant from the majority of instances in each view.

- **Class outlier**: Instances whose neighborhood structures differ remarkably between views.

- **Attribute-Class outlier**: Instances that exhibit both types of attribute outlier and class outlier.

These neighborhood-based multi-view outlier detection methods (Ji et al., 2019; Wang et al., 2023; Cheng et al., 2021) have made significant progress. However, they still face two major challenges, making this task fundamentally difficult.

- **Outlier propagation contaminates learning.** Most existing methods assume that all samples and neighbors are reliable during neighborhood construction and cross-view consensus learning (Chao et al., 2025). As a result, erroneous neighborhoods are repeatedly reinforced, affecting performance.

- **Scale discrepancy across views.** Multi-view data inherently suffer from numerical scale discrepancy (differences in feature magnitudes across views). Normalization in the input space can largely alleviate numerical scale inconsistency, yet inconsistency persists after encoding, since the latent representations may undergo nonlinear transformations with view-specific scaling. These discrepancies distort neighborhood geometry and misalign cross-view structures, ultimately hindering reliable consensus learning.

To tackle the above challenges, we propose RNAMOD (Reliable Neighborhood-Aware Multi-View Outlier Detection), a reliability-driven MVOD framework that suppresses outlier propagation, selects more reliable neighbors, and mitigates inter-view scale discrepancy by aligning geometric directions. Our contributions are summarized as follows:

1. We introduce the concept of reliability that dynamically updates the reliability of each sample in every training iteration, suppressing the influence of anomalous samples on the learning of normal data structure, and thus preventing outlier propagation.

2. We design a reliable neighborhood construction strategy that avoids the selection of outliers, thus enhancing the quality of the neighborhood structure and mitigating the outlier propagation.

3. We design a leave-one-out directional consensus learning mechanism that alleviates self-reinforcement effects and mitigates inter-view scale discrepancy by aligning neighborhood directions.

4. We incorporate a dynamic view weighting scheme that captures the global structural quality of each view, suppressing noisy or low-quality views during training.

## 2. Related Works

### 2.1. Clustering-Based Outlier Detection

Outlier detection has achieved significant success in the past two decades (Zhang et al., 2024; He et al., 2003; Sathe & Aggarwal, 2016; Zong et al., 2018; Guo & Zhu, 2018; Janeja & Palanisamy, 2013), but most methods have been designed for single-view data, which limits their ability to capture the various aspects of the data and results in poor generalization. With the advancement of multi-view learning, multi-view outlier detection has emerged as an important task (Gao et al., 2011) and has seen considerable progress, especially clustering-based outlier detection methods. For example, the HOAD method (Gao et al., 2011) simultaneously performs spectral clustering on the similarity graphs derived from two views while imposing a consistency constraint. This approach first examines "inconsistent behaviors" in multi-view data and then utilizes the results of multi-view spectral clustering to predict the presence of inconsistency in an instance. This is a pioneering work in multi-view outlier detection.

Similarly, APOD (Marcos Alvarez et al., 2013) applies affinity propagation clustering separately in multiple views to obtain affinity vectors for each object, subsequently detecting and identifying outliers by comparing these cross-view affinity vectors. However, these methods employ relatively simple definitions of an outlier and are less effective in detecting outliers. DMOD and CRMOD (Zhao et al., 2017) respectively, proposed the use of k-means and consensus learning to unify clustering indicators across different views. Similarly, MLRA (Li et al., 2015a) and LDSR (Li et al., 2018) explored subspace clustering. However, all of these approaches are based on clustering for outlier detection, which typically assumes that the data set inherently exhibits clustering characteristics, thus limiting the generalizability of the models.

### 2.2. Neighborhood-Based Outlier Detection

Recent methods have adopted neighborhood based approaches for outlier detection (Sheng et al., 2019; Ji et al., 2019; Hu et al., 2024; Cheng et al., 2021; Wang et al., 2023; 2024). Unlike clustering-based methods, these approaches do not require explicit cluster structures and instead identify anomalies based on the set of nearest neighbors of each

instance. In particular, an instance classified as anomalous under clustering assumptions will typically also be anomalous under neighborhood-based definitions.

*MODDIS* (Ji et al., 2019) was the first work to introduce representation learning into multi-view outlier detection by projecting each view and cross-view representations into a shared latent space via a neural network, followed by k-nearest neighbor-based outlier identification. *NC-MOD* (Cheng et al., 2021) follows a similar strategy, employing autoencoders and constructing a kNN consensus matrix in the latent space. *SRLSP* (Wang et al., 2023) integrates self-representation and consensus learning to capture multi-view structure, without explicit dimensionality reduction. *MODGD* (Hu et al., 2024) further enhances detection by fusing neighborhood graphs, while *RCPMOD* (Wang et al., 2024) introduces a contrastive learning framework tailored for multi-view outlier detection.

In addition, several methods detect outliers by measuring the difference between learned representations (Lin et al., 2025). However, most existing approaches implicitly assume that the neighbors of a sample provide reliable semantic information. There is a flaw in this assumption: outliers should not be treated as valid participants when modeling multi-view structures. In addition, when measuring cross-view inconsistencies, differences in scale often cause certain views to dominate the training.

To address these issues, we propose RNAMOD, whose main idea is to leverage reliability cues to suppress the adverse influence of outliers and ensure that the learned multi-view structures are primarily shaped by normal data.

## 3. Method

### 3.1. Setup

The multi-view data set is formally defined as $\mathbf{X} = \{\mathbf{X}^1, \mathbf{X}^2, \ldots, \mathbf{X}^V\}$, where $\mathbf{X}^v \in \mathbb{R}^{d_v \times N}$ denotes the feature matrix of the $v$-th view, $d_v$ represents the dimension of the $v$-th view and $N$ indicates the number of aligned cross-view samples. Specifically, the multi-view feature matrix for the $v$-th view is defined as $\mathbf{X}^v = \{x_1^v, x_2^v, \ldots, x_N^v\}$, where $\mathbf{x}_i^v \in \mathbb{R}^{d_v}$ denotes the $i$-th instance in the $v$-th view with dimension $d_v$.

The framework of the proposed RNAMOD method is illustrated in Figure 2. It illustrates two outlier-contaminated learning scenarios, denoted as C1 and C2, which are described in detail in Section 3.2. We first project the data into a latent space via autoencoders and construct reliable neighborhoods to prevent outlier propagation. A leave-one-out directional consensus module is further employed to align neighborhood directions, thereby mitigating inter-view scale discrepancy and self-reinforcement effects.

### 3.2. Motivation

Multi-view outlier detection aims to recover the underlying structure of normal data from outlier-contaminated datasets. However, outliers can corrupt cross-view consensus learning if they are incorporated into the construction of neighborhood structures, i.e. the set of neighbors. To prevent such outlier propagation, we formalize two necessary conditions to satisfy:

**C1** Outliers should not be selected as neighbors of other samples, as projection into the latent space may place them close to normal instances.

**C2** Outliers should not participate in cross view neighborhood alignment, since class outliers' neighborhood structures are inconsistent across views, and enforcing alignment would propagate noise. Moreover, attribute outliers are far from most instances, making their neighborhood structures meaningless.

Recent work such as RCPMOD (Wang et al., 2024) mitigates outlier bias through contrastive learning and regularization, but constructs KNN graphs for all samples and applies alignment uniformly, thus violating **C1** and **C2**.

In unsupervised settings, true outliers are unknown; however, their outlier scores allow us to estimate reliability. We define three types of reliability scores and use them to prevent outlier propagation and model view quality.

### 3.3. Reliable Neighborhood Construction

Following recent advances in deep unsupervised multi-view outlier detection (Ji et al., 2019; Wang et al., 2024; Cheng et al., 2021), we adopt view-specific autoencoders to learn latent representations. The reconstruction loss is defined as:

$$\mathcal{L}_{rec}^v = \sum_{i=1}^N \|x_i^v - g^v(f^v(x_i^v))\|_2^2, \qquad (1)$$

where $f^v(\cdot)$ and $g^v(\cdot)$ denote the encoder and the decoder of the $v$th view, respectively.

A key observation is that attribute outliers, being globally corrupted across all views, cannot be faithfully reconstructed by the autoencoder. Therefore, the reconstruction error can be considered as an inverse indicator of the confidence that a sample is not an attribute outlier under the current parameters $\Theta$. Motivated by this observation, we define the sample reliability as follows.

$$\phi_{i,v} = 2\sigma\big(-\alpha\|x_i^v - \hat{x}_i^v\|_2\big), \quad \phi_i = \frac{1}{V}\sum_{v=1}^V \phi_{i,v}, \quad (2)$$

where $\sigma(\cdot)$ denotes the sigmoid function and $\alpha > 0$ is a hyperparameter. By design, $\phi_i \approx 0$ for attribute outliers and $\phi_i \approx 1$ for others under $\Theta$.

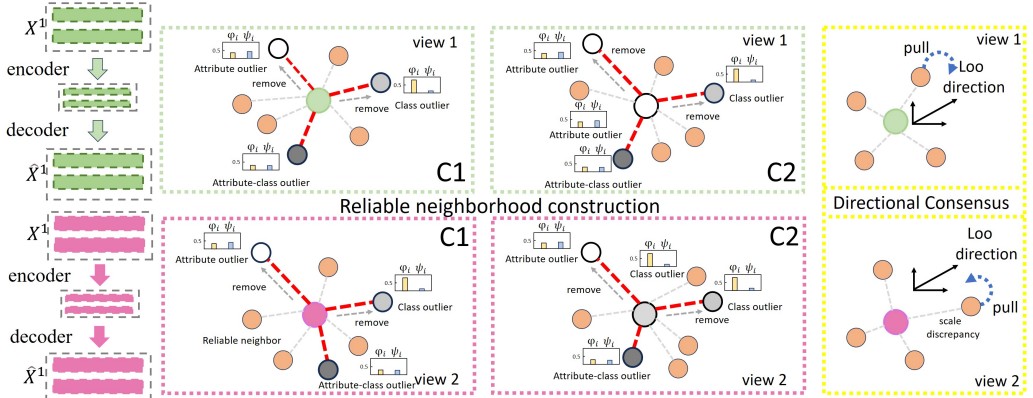

*Figure 2.* Overall framework of the proposed method RNAMOD. For each view, an autoencoder is trained to project the data into a latent space for representation learning. C1 and C2 illustrate two typical cases of outlier propagation that contaminate neighborhood consistency. The proposed reliable neighborhood construction (left) dynamically estimates reliability and removes unreliable outlier samples, improving the quality of neighborhood structures. The leave-one-out directional consensus (right) aligns neighborhood directions across views, thereby mitigating inter-view scale discrepancy and preventing self-reinforcement during cross-view learning.

Similarly, we define consistency reliability as the confidence that the sample is not a class outlier under $\Theta$:

$$\psi_i = \left(\frac{1}{V(V-1)} \sum_{v=1}^{V} \sum_{u \neq v} \frac{|\mathcal{N}_i^v \cap \mathcal{N}_i^u|}{K}\right)^p \qquad (3)$$

where $\mathcal{N}_i^v$ denotes the set of neighbors of the $i$th instance in the $v$th view. Setting $p < 1$ drives $\psi_i$ closer to 1, which prevents them from becoming excessively small and diminishing the contribution of other components.

Equipped with $\phi_i$ and $\psi_i$, we construct a reliable neighborhood construction to represent the neighborhood structure. The main motivation is to ensure that outliers are not treated as trustworthy neighbors during neighborhood construction. Specifically, we first define the neighborhood score in $v$th view as:

$$s_{ij}^v = \frac{\|h_i^v - h_j^v\|_2}{\phi_j \psi_j + \varepsilon}, \qquad (4)$$

and the final neighborhood structure as:

$$\mathcal{N}_i^v = \text{KNN}\left(s_{ij}^v\right). \quad j \in 1, 2, ..., N \qquad (5)$$

where $\varepsilon > 0$ is a small constant introduced to prevent division by zero, $K > 0$ controls the size of the neighborhood, and $h_i^v = f^v(x_i^v)$ denotes the latent representation of the $i$th instance in the $v$th view. Specifically, we select the K-smallest instances.

For an attribute outlier $j$, the sample reliability satisfies $\phi_j^v \to 0$ across all views, which causes $s_{ij}^v \to \infty$ for $j$. For a class outlier $l$, although $\phi_l \to 1$, its class reliability $\psi_l \to 0$ due to cross-view inconsistency. As for attribute-class outlier q, both $\phi_q^v \to 0$ (in perturbed views) and $\psi_q \to 0$, ensuring that such samples are not selected as neighbors by others.

Thus, the proposed reliable neighborhood construction mechanism resolves **C1** by design.

### 3.4. Leave-One-Out Directional Consensus

Even with reliable neighborhood construction, aligning neighborhood structures across views is still required to measure their difference. However, since multi-view representations are produced by independent nonlinear encoders, their latent spaces often exhibit view-specific numerical scale discrepancy. Under such a discrepancy, enforcing representation or distance-level consistency typically introduces view bias and hinders the integration of complementary neighborhood information. To address this issue, we align the directions of neighborhood edges, which are more invariant across views. Moreover, we design a leave-one-out directional consensus scheme to avoid self-reinforcement during alignment. Accordingly, we define the following three concepts:

**Reliable edge set.** Let $\mathcal{N}_i^v$ be the reliable neighbor set of the $i$th sample in the $v$th view. The corresponding reliable edge set is denoted as $\mathcal{E}_i^v = \{(i,j) | j \in \mathcal{N}_i^v\}$.

**Directional consensus.** For each edge $(i,j) \in \mathcal{E}_i^v$, the view-specific direction is defined as

$$d_{ij}^v = h_i^v - h_j^v, \qquad (6)$$

and the corresponding leave-one-out consensus is

$$d_{ij}^{-v} = \frac{1}{V-1} \sum_{u \neq v} d_{ij}^u. \qquad (7)$$

**View weight.** Let $\pi = \{\pi_1, \pi_2, ..., \pi_V\}$ denote the weights of different views.

Self-reinforcement occurs when a view validates its own biased structure during alignment, and the leave-one-out strategy eliminates this bias by excluding the view itself from the consensus estimation. Based on directional consensus, we define the directional alignment loss for the $i$th sample in the $v$th view as:

$$\mathcal{L}_{\text{ang,i}}^v = \frac{1}{|\mathcal{E}_i^v|} \sum_{(i,j)\in\mathcal{E}_i^v} \left(1 - \cos(d_{ij}^v, d_{ij}^{-v})\right). \quad (8)$$

which minimizes angular deviations among reliable edges to achieve affine-invariant and noise-robust neighborhood consensus.

On the other hand, the loss $\mathcal{L}_{\text{ang,i}}^v$ reflects how much a sample in the $v$th view deviates from the other views. By aggregating it over all samples, i.e. $\sum_{i=1}^N \mathcal{L}_{\text{ang,i}}^v$ we can quantify the overall deviation of the $v$th view from the multi-view consensus, this accurately reflects the relative importance of each view. Formally, the quality weight of the $v$th view is defined as:

$$\pi_v = \frac{\exp(-E_v/\tau_\pi)}{\sum_{u=1}^V \exp(-E_u/\tau_\pi)}, \qquad E_v = \sum_{i=1}^N \mathcal{L}_{\text{ang,i}}^v, \quad (9)$$

where $\tau_\pi$ prevents early collapse for weak views.

### 3.5. Overall Loss Function

The overall training loss function integrates two complementary components: (1) **view-specific reconstruction loss**, which ensures faithful data recovery in each view; (2) **loss of directional consensus**, which enforces consistent neighborhood directions in all views. Importantly, it is crucial to effectively prevent outlier propagation and mitigate scale discrepancy across views. In particular, to satisfy **C2**, we assign reliability $\phi_j\psi_j$, ensuring that only reliable samples participate in consensus learning. Moreover, since views with higher weight should produce more accurate representations, we assign larger view weights to these views, allowing their reliable structures to dominate the optimization process. Formally, the total loss function is defined as follows:

$$\mathcal{L} = \sum_{v=1}^V \pi_v \mathcal{L}_{rec}^v + \lambda \sum_{v=1}^V \sum_{i=1}^N \phi_i\psi_i\mathcal{L}_{\text{ang,i}}^v, \quad (10)$$

$\mathcal{L}_{\text{rec}}^v$ is the reconstruction loss for the $v$th view, $\mathcal{L}_{\text{ang,i}}^v$ is the leave-one-out directional consensus loss, and $\lambda$ balances the importance.

### 3.6. Outlier Scoring

We design two complementary outlier scores to characterize three types of outliers in multi-view data. The attribute outlier score evaluates whether a sample fails to be well reconstructed by the underlying autoencoder in any view:

$$S_i^{\text{attr}} = \sum_{v=1}^V \|x_i^v - \hat{x}_i^v\|_2^2, \quad (11)$$

A high $S_i^{\text{attr}}$ indicates that $x_i$ is more likely to be an attribute outlier.

To detect class outliers, we further define a class outlier score by measuring the disagreement between each reliable neighborhood and its leave-one-out consensus:

$$S_i^{\text{class}} = \frac{1}{V} \sum_{v=1}^V \left(\mathcal{L}_{\text{ang},i}^v + \mathcal{L}_{\text{dis},i}^v\right), \quad (12)$$

where $\mathcal{L}_{\text{ang},i}^v$ and $\mathcal{L}_{\text{dis},i}^v$ denote the directional and distance disagreements between the $v$th view and the consensus, respectively. For a reliable edge $(i,j) \in \mathcal{E}_i^v$, the distance term is defined as

$$\mathcal{L}_{\text{dis},i}^v = \frac{1}{|\mathcal{E}_i^v|} \sum_{(i,j)\in\mathcal{E}_i^v} \left| \|d_{ij}^v\|_2 - \bar{d}_{ij}^{-v} \right|, \quad (13)$$

where $\bar{d}_{ij}^{-v}$ is the leave-one-out consensus distance, and

$$\bar{d}_{ij}^{-v} = \frac{1}{V-1} \sum_{u\neq v} \|d_{ij}^u\|_2$$

is the average cross-view edge length. During training, only the angular term is used to ensure scale invariance and stable optimization, while the distance term is incorporated during scoring to explicitly capture magnitude information that is not represented by directional consistency.

Finally, we combine the two scores to obtain the overall outlier score:

$$S_i = \frac{S_i^{\text{attr}} - \mu_{\text{attr}}}{\sigma_{\text{attr}}} + \frac{S_i^{\text{class}} - \mu_{\text{class}}}{\sigma_{\text{class}}}, \quad (14)$$

where $\mu_{\text{attr}}, \sigma_{\text{attr}}$ and $\mu_{\text{class}}, \sigma_{\text{class}}$ denote the mean and standard deviation of the attribute and class outlier scores, respectively. A higher $S_i$ indicates that the sample is more likely to be an outlier. In particular, view weights $\pi$ are excluded from the outlier score, as they only govern representation learning.

## 4. Optimization

We design an alternating optimization to jointly update the parameters of RNAMOD. Let $t$ denote the current training epoch, and $\phi_i^t$ and $\psi_i^t$ represent the sample and consistency reliability computed under the parameters $\Theta^t$. The deep autoencoders are first pre-trained to obtain well-initialized parameters. Then, the overall optimization process alternates

between reliability, view weight, neighborhood construction, and network updating, as summarized below:

**(1) Update $\phi_i^t$ and $\psi_i^t$.** Compute $\phi_i^t$ and $\psi_i^t$ using Eqs. 2 and 3 under the parameters $\Theta^{t-1}$. Specifically, when $t = 1$, conventional $k$-nearest neighbors in the latent space are used to initialize the neighborhood structures $\mathcal{N}_i^v$ for computing $\psi_i$.

**(2) Update view weights $\pi^t$.** Compute $\pi_v^t$ via Eq. 9 under $\Theta^{t-1}$. In the first iteration ($t = 1$), $\pi$ is uniformly initialized as:

$$\pi^1 = \left\{ \frac{1}{V}, \frac{1}{V}, \dots, \frac{1}{V} \right\}. \tag{15}$$

**(3) Update the reliable neighborhood structure.** Fixing $\phi_i^t$ and $\psi_i^t$, construct the reliable neighborhood structure $\{\mathcal{N}_i^{v,t}\}$ for each view by filtering out unreliable neighbors, and derive the corresponding edge sets $\{\mathcal{E}_i^{v,t}\}$.

**(4) Update the network parameters.** With $\phi_i^t$, $\psi_i^t$ and $\pi$ fixed, update the network parameters $\Theta^{t+1}$ by minimizing the total loss in Eq. 10.

This alternating process continues until convergence.

# 5. Experiments

## 5.1. Setup

Six real-world datasets from the work (Li et al., 2015b) are used as benchmark datasets. These datasets are multi-view, multi-class, and originally outlier-free. In addition, we separately construct a synthetic dataset to demonstrate the effectiveness of the proposed reliable neighborhood construction. This dataset is randomly generated, outlier-free, and designed without any explicit clustering structure. The statistics of all datasets are summarized in Table 2, where $Dim$ denotes the dimensionality of each view, and the synthetic dataset is denoted as Synthetic.

Following prior work (Hu et al., 2024; Ji et al., 2019; Wang et al., 2024), we simulate the three types of outliers, using the following strategies: (1) *Attribute outlier*: Randomly selected instances are replaced by random values in all views; (2) *Class outlier*: Feature representations of instance pairs from different classes are randomly swapped in half of the views ($\frac{V}{2}$); (3) *Attribute-class outlier*: Selected instances are simultaneously subjected to attribute perturbations in half of the views and class-level feature exchanges in the remaining views.

To evaluate the performance of outlier detection in real-world datasets with varying outlier ratios, we use the outlier ratios listed in Table 1, where `id` denotes the configuration index and $\rho_1$, $\rho_2$ and $\rho_3$ represent the proportions of attribute, class, and attribute-class outliers, respectively. Each dataset is denoted by **dataset abbreviation + configuration**

*Table 1.* Outlier Ratio Settings.

| id | 1 | 2 | 3 | 4 | 5 | 6 |
|----|------|------|------|------|------|------|
| $\rho_1$ | 0.02 | 0.02 | 0.05 | 0.05 | 0.08 | 0.08 |
| $\rho_2$ | 0.05 | 0.08 | 0.02 | 0.08 | 0.02 | 0.05 |
| $\rho_3$ | 0.08 | 0.05 | 0.08 | 0.02 | 0.05 | 0.02 |

*Table 2.* Datasets and their statistics.

| Dataset | # Nums | #Dim |
|---------|--------|------|
| Synthetic | 240 | 8:8 |
| 100Leaves | 1000 | 64:64:64 |
| BBCSport | 544 | 3183:3203 |
| Reuters | 1200 | 21531:24893:34279:15506:11519 |
| Caltech101 | 9144 | 48:40:254:1984:512:928 |
| NUSWIDEOBJ | 30000 | 65:226:145:74:129 |
| CiteSeer | 3312 | 3312:3703 |

**index**. Each method-dataset combination was executed for five independent runs and the mean AUC is reported in all runs. For the synthetic dataset, we use a fixed outlier ratio of 0.05-0.05-0.05. It is used only to illustrate the effectiveness of reliable neighborhood construction rather than to report AUC.

Seven representative MVOD methods are selected as baselines: APOD (Marcos Alvarez et al., 2013), MODDIS (Ji et al., 2019), NCMOD (Cheng et al., 2021), SRLSP (Wang et al., 2023), MODGD (Hu et al., 2024), RCPMOD (Wang et al., 2024) and LRTDM (Lin et al., 2025). They can be categorized as follows:

**Clustering-based methods**. *APOD* (Marcos Alvarez et al., 2013) performs affinity propagation clustering in each view and detects outliers through inconsistency in cluster of different views. Although simple, its effectiveness relies heavily on clear cluster structures.

**Neighborhood-based methods**. *MODDIS* (Ji et al., 2019) and *NCMOD* (Cheng et al., 2021) learn view-specific latent spaces and utilize neighborhood discrepancy to identify multi-type outliers. *SRLSP* (Wang et al., 2023) enhances local similarity learning with a consensus graph. *MODGD* (Hu et al., 2024) further incorporates structured sparsity to denoise multiple neighborhood graphs. *RCPMOD* (Wang et al., 2024) exploits regularized contrastive learning to improve robustness against view inconsistency and missing views.

In addition, *LRTDM* (Lin et al., 2025) is a different approach. It performs low-rank Tucker decomposition on the self-expressive tensor and uses meta-learning to fuse view-specific latent representations, detecting class outliers based on measuring the inconsistency between each view's self-expressive matrix and a meta-learned consensus rather than neighborhood structures.

*Table 3.* AUC performance on 100Leaves, Caltech101 and NUSWIDEOBJ, with the best result in each column in bold and the second best underlined.

| Method | 100Leaves | | | | | | Caltech101 | | | | | | NUSWIDEOBJ | | | | | |
|---|---|---|---|---|---|---|---|---|---|---|---|---|---|---|---|---|---|---|
| ratio | 1 | 2 | 3 | 4 | 5 | 6 | 1 | 2 | 3 | 4 | 5 | 6 | 1 | 2 | 3 | 4 | 5 | 6 |
| APOD | 0.791 | 0.586 | 0.808 | 0.673 | 0.712 | 0.714 | 0.689 | 0.605 | 0.677 | 0.730 | 0.727 | 0.776 | 0.829 | 0.635 | 0.827 | 0.712 | 0.805 | 0.777 |
| MODDIS | 0.779 | 0.679 | 0.879 | 0.776 | 0.792 | 0.794 | 0.696 | 0.671 | 0.763 | 0.693 | 0.743 | 0.775 | 0.841 | 0.707 | 0.876 | 0.728 | 0.821 | 0.776 |
| NCMOD | 0.789 | 0.803 | 0.908 | 0.795 | 0.857 | 0.820 | 0.773 | 0.704 | 0.769 | 0.720 | 0.830 | 0.772 | 0.825 | 0.691 | 0.882 | 0.709 | 0.832 | 0.773 |
| SRLSP | 0.832 | 0.808 | 0.879 | 0.822 | 0.881 | 0.859 | 0.786 | 0.813 | 0.856 | 0.721 | 0.851 | 0.787 | **0.888** | 0.728 | 0.857 | 0.739 | 0.858 | 0.796 |
| MODGD | 0.810 | 0.743 | 0.892 | 0.790 | 0.906 | 0.856 | 0.838 | 0.757 | 0.921 | 0.707 | 0.807 | 0.823 | 0.822 | 0.711 | 0.948 | 0.741 | 0.959 | 0.824 |
| LRTDM | 0.847 | 0.759 | 0.903 | 0.804 | 0.895 | 0.887 | 0.841 | 0.834 | 0.885 | 0.857 | 0.840 | 0.852 | 0.836 | 0.812 | 0.964 | 0.802 | 0.941 | 0.851 |
| RCPMOD | **0.875** | **0.837** | 0.948 | 0.830 | 0.863 | **0.921** | 0.862 | 0.887 | 0.899 | 0.854 | 0.880 | 0.885 | 0.834 | 0.829 | 0.960 | **0.818** | 0.956 | 0.882 |
| **Ours** | 0.856 | 0.767 | **0.952** | **0.846** | **0.945** | **0.921** | **0.895** | **0.908** | **0.907** | **0.864** | **0.902** | **0.909** | 0.857 | **0.887** | **0.973** | 0.807 | **0.971** | **0.923** |

*Table 4.* AUC performance on BBCSport, Reuters and CiteSeer, with the best result in each column in bold and the second best underlined.

| Method | BBCSport | | | | | | Reuters | | | | | | CiteSeer | | | | | |
|---|---|---|---|---|---|---|---|---|---|---|---|---|---|---|---|---|---|---|
| ratio | 1 | 2 | 3 | 4 | 5 | 6 | 1 | 2 | 3 | 4 | 5 | 6 | 1 | 2 | 3 | 4 | 5 | 6 |
| APOD | 0.781 | 0.695 | 0.720 | 0.702 | 0.710 | 0.715 | 0.688 | 0.582 | 0.758 | 0.596 | 0.775 | 0.670 | 0.666 | 0.605 | 0.748 | 0.598 | 0.742 | 0.664 |
| MODDIS | 0.802 | 0.720 | 0.760 | 0.728 | 0.754 | 0.744 | 0.722 | 0.605 | 0.798 | 0.633 | 0.795 | 0.695 | 0.714 | 0.647 | 0.782 | 0.633 | 0.780 | 0.700 |
| NCMOD | 0.822 | 0.747 | 0.790 | 0.750 | 0.777 | 0.770 | 0.738 | 0.625 | 0.830 | 0.652 | 0.820 | 0.715 | 0.730 | 0.662 | 0.810 | 0.646 | 0.803 | 0.715 |
| SRLSP | 0.845 | 0.787 | 0.828 | 0.804 | 0.838 | 0.812 | 0.809 | 0.708 | 0.925 | 0.734 | 0.912 | 0.832 | 0.848 | 0.720 | 0.928 | 0.720 | 0.918 | 0.834 |
| MODGD | 0.830 | 0.770 | 0.822 | 0.792 | 0.880 | 0.806 | 0.793 | 0.636 | 0.933 | **0.742** | 0.923 | 0.846 | 0.851 | 0.750 | 0.929 | 0.726 | 0.932 | 0.842 |
| LRTDM | 0.833 | 0.764 | 0.815 | 0.824 | 0.842 | 0.870 | 0.756 | 0.650 | 0.878 | 0.703 | 0.876 | 0.794 | 0.835 | 0.723 | 0.916 | 0.721 | 0.925 | 0.826 |
| RCPMOD | 0.842 | 0.772 | 0.823 | 0.900 | 0.908 | 0.920 | 0.785 | 0.700 | 0.918 | 0.728 | 0.934 | 0.852 | 0.865 | 0.700 | 0.934 | 0.728 | 0.894 | 0.814 |
| **Ours** | **0.870** | **0.810** | **0.875** | **0.924** | **0.952** | **0.952** | **0.838** | **0.770** | **0.944** | 0.740 | **0.958** | **0.867** | **0.903** | **0.758** | **0.947** | **0.748** | **0.964** | **0.908** |

## 5.2. Results and Discussion

Two sets of experiments are conducted. The first evaluates the mitigation of outlier propagation under different neighborhood construction strategies. The second reports the main AUC results, comparing our method with the baselines.

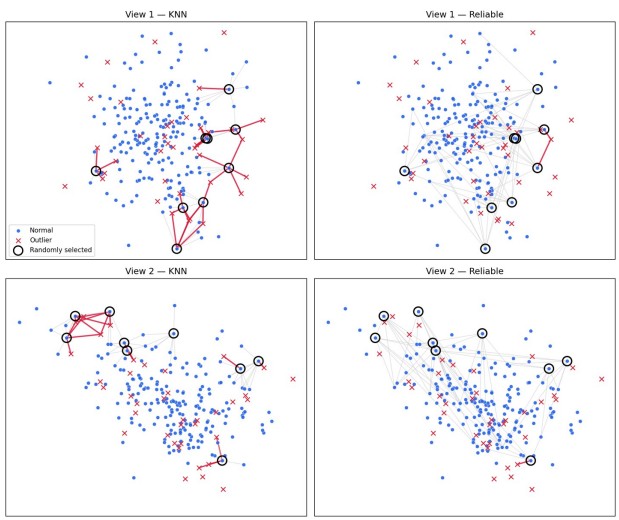

*Figure 3.* Qualitative comparison of neighborhood construction on the synthetic dataset.

Figure 3 illustrates the neighborhood construction results in the synthetic dataset. KNN denotes the conventional distance-based neighborhood construction, while Reliable represents our proposed reliable neighborhood construction. The data dimensions are projected to two dimensions us-

ing an autoencoder for visualization. Red lines indicate neighbor connections involving outliers. The red lines are greatly reduced, indicating that our reliable neighborhood construction filters out outliers and produces a higher-quality neighborhood structure.

Table 3 and Table 4 summarize the AUC performance of all compared methods on real-world datasets. Overall, our proposed RNAMOD consistently achieves the best or second-best performance in the majority of benchmark datasets, the following are some analyses.

First, APOD achieves the lowest performance in almost all benchmark datasets. This is because it is constrained by a strong clustering assumption: when view misalignment occurs within data belonging to the same category, clustering-based methods are no longer effective, which leads to a significant degradation in its performance.

Second, neighborhood-based methods such as SRLSP, MODGD, and RCPMOD perform well in many cases. In particular, SRLSP achieves the second-best performance on the BBCSport dataset, while RCPMOD achieves the second-best results in several other settings. However, SRLSP lacks an explicit mechanism to handle outlier propagation; MODGD performs only static graph denoising with fixed neighborhood structures; and RCPMOD merely restricts propagation via an outlier-aware contrastive loss without modeling reliability. In contrast, RNAMOD adaptively refines both neighborhood construction and cross-view consensus through reliability estimation, leading to more stable detection performance. LRTDM relies on global low-rank reconstruction, making it sensitive to view-specific scal-

*Table 5.* Ablation study of RNAMOD on six datasets.

| Variant | Components | | | | | AUC | | | | | |
|---|---|---|---|---|---|---|---|---|---|---|---|
| | SR | CR | AL | LOO | VW | 100Leaves | Caltech101 | NUSWIDEOBJ | BBCSport | Reuters | CiteSeer |
| (A) | × | × | × | × | × | 0.802 | 0.791 | 0.842 | 0.824 | 0.801 | 0.828 |
| (B) | ✓ | ✓ | ✓ | × | × | 0.831 | 0.891 | 0.884 | 0.876 | 0.838 | 0.854 |
| (C) | ✓ | ✓ | ✓ | ✓ | × | 0.826 | 0.884 | 0.892 | 0.874 | 0.842 | 0.850 |
| (D) | × | × | ✓ | ✓ | ✓ | 0.819 | 0.873 | 0.850 | 0.862 | 0.816 | 0.833 |
| (E) | ✓ | × | ✓ | ✓ | ✓ | 0.824 | 0.879 | 0.861 | 0.868 | 0.823 | 0.841 |
| (F) | × | ✓ | ✓ | ✓ | ✓ | 0.826 | 0.872 | 0.885 | 0.871 | 0.828 | 0.837 |
| (G) | ✓ | ✓ | ✓ | ✓ | ✓ | **0.848** | **0.898** | **0.911** | **0.897** | **0.853** | **0.871** |

ing discrepancy, whereas RNAMOD explicitly aligns directional patterns to mitigate such scale effects.

## 5.3. Parameter Sensitivity Analysis

To evaluate the impact of different parameters on the proposed RNAMOD, Figure 4 illustrates the performance with varying $\alpha$, $p$, $k$ and $\lambda$ in six real-world datasets. The default values are set as $\alpha = 0.5$, $p = 0.25$, $k = 10$ and $\lambda = 1$.

**Impact of $\alpha$.** As shown in the upper-left subplot of Figure 4, the performance remains stable at most values of $\alpha$, except when $\alpha = 0$ or $\alpha = 1$. When $\alpha = 0$, the sample reliability is removed, causing unreliable samples to participate in neighborhood construction and amplifying outlier propagation. In contrast, when $\alpha = 1$, the reconstruction error is overly emphasized, making the model sensitive to inter-view density and scale discrepancy.

**Impact of $p$.** As shown in the upper-right subplot of Fig. 4, the optimal value is obtained when $p = 0.25$. This is because even among normal samples, there are noticeable structural differences between views. A smaller $p$ increases the value $\psi_i$, preventing them from excessively suppressing other components. However, if $p$ becomes too small, the differences between $\psi_i$ shrink, weakening its discriminative ability.

**Impact of $k$ and $\lambda$.** As shown in the lower-left and lower-right subplots of Figure 4, the performance exhibits a bell-shaped trend with respect to both $k$ and $\lambda$. The best results are obtained when $k = 10$ and $\lambda = 1$, where the neighborhood structure is sufficiently informative and the regularization achieves an appropriate balance between reconstruction fidelity and directional alignment.

## 5.4. Ablation Study

We conducted an ablation study on five modules, which are described as follows:

**Sample Reliability (SR)** Removal of this module sets the sample reliability to $\phi_i = 1$ for all samples.

**Consistency Reliability (CR)** Removal of this module sets

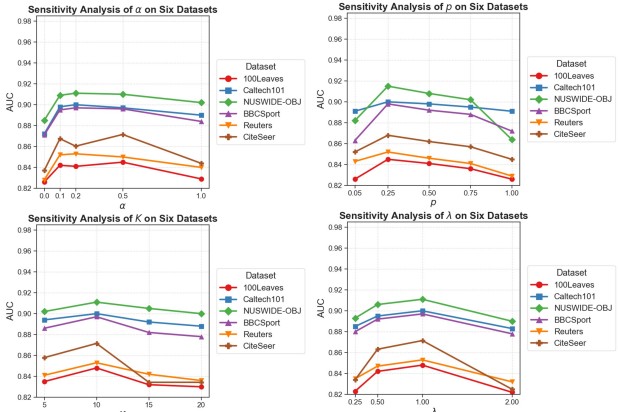

*Figure 4.* Sensitivity analysis over $\alpha$, $p$, $k$ and $\lambda$.

the consistency reliability at $\psi_i = 1$ for all samples.

**Alignment Loss (AL)** Removing this module leaves the objective with only the reconstruction loss. In this case, both SR and CR become ineffective, since they no longer affect the objective.

**Leave-One-Out Consensus (LOO)** Removal of this module replaces the directional leave-one-out consensus with a full consensus that includes the current view.

**View Weighting (VW)** Removal of this module fixes the view weights to $\frac{1}{V}$ for all views. Since VW depends on LOO, when LOO is removed, VW becomes inactive.

The ablation results are reported in Table 5, where ✓ and × denote the corresponding modules enabled and disabled, respectively. It is evident that each module contributes positively to the overall outlier detection performance, as observed in variants (A)–(F). In particular, SR and CR provide comparable improvements from (E), (F). In contrast, the impact of LOO and VW is relatively smaller than that of SR and CR, as can be seen from variants (B), (C), (E), and (F), suggesting that mitigating outlier propagation has a more substantial effect than simply alleviating the scale discrepancy across views.

## 6. Conclusion

In this paper, we proposed **RNAMOD**, a reliable neighborhood-aware framework for multi-view outlier detection. By modeling reliability and aligning neighborhood directions across views, **RNAMOD** effectively suppresses outlier propagation and mitigates inter-view scale discrepancy. Extensive experiments on both synthetic and real-world datasets demonstrate its superior performance.

## Acknowledgements

This work is supported in part by the National Natural Science Foundation of China(No.62276079), and the Special Funding Program of Shandong Taishan Scholars Project.

## Impact Statement

This paper presents work whose goal is to advance the field of Machine Learning. There are many potential societal consequences of our work, none which we feel must be specifically highlighted here.

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

| Variant | Components | | | | | AUC | | | | | |
|---|---|---|---|---|---|---|---|---|---|---|---|
| | AB | DB | AS | DS | FS | 100Leaves | Caltech101 | NUSWIDEOBJ | BBCSport | Reuters | CiteSeer |
| (A) | × | ✓ | ✓ | ✓ | ✓ | 0.795 | 0.824 | 0.826 | 0.813 | 0.810 | 0.823 |
| (B) | ✓ | ✓ | ✓ | ✓ | ✓ | 0.831 | 0.891 | 0.902 | 0.871 | 0.841 | 0.862 |
| (C) | ✓ | × | × | ✓ | × | 0.813 | 0.840 | 0.841 | 0.852 | 0.812 | 0.821 |
| (D) | ✓ | × | ✓ | × | × | 0.793 | 0.822 | 0.814 | 0.803 | 0.782 | 0.814 |
| (E) | ✓ | × | ✓ | ✓ | ✓ | **0.848** | **0.898** | **0.911** | **0.897** | **0.853** | **0.871** |

*Table 6.* Supplementary ablation study of RNAMOD on six datasets.

## A. Time Complexity

The overall training time complexity per iteration is dominated by two primary computational bottlenecks: reliable neighborhood construction and directional consensus calculation.

**Reliable Neighborhood Construction:** This involves computing neighborhood scores $s_{ij}^v$ (Equation 4) and selecting the $K$ nearest neighbors for each of the $N$ samples across $V$ views using naive pairwise distance computation, resulting in a time complexity of $\mathcal{O}(VN^2 D_h)$, where $D_h$ denotes the dimensionality of the latent space.

**Directional Consensus Calculation:** This module calculates the directional consensus loss $\mathcal{L}_{ang,i}^v$ and total loss (Eq. 10). For each of the $N$ samples, we iterate through its $K$ neighbors across $V$ views to compute the leave-one-out consensus direction $d_{ij}^{-v}$ and the cosine similarity loss. This leads to a complexity of $\mathcal{O}(V^2 NKD_h)$, which is linear with respect to $K$.

The total time complexity per training epoch is thus:

$$\mathcal{O}(VN^2 D_h + V^2 NKD_h) \tag{16}$$

Given that $K$ and $D_h$ are typically small constants independent of $N$, the overall time complexity is $\mathcal{O}(N^2)$, which is comparable to that of prior neighborhood-based methods (Wang et al., 2024; Hu et al., 2024).

In addition, for larger-scale datasets, the runtime of our method can be further reduced by decreasing the frequency of reliable neighborhood construction, i.e., performing it once every several epochs rather than at every epoch. We conduct experiments on the six real-world datasets used in the main paper to evaluate how the AUC varies under different update intervals, and report the results in Fig. 5. As can be seen, although the update frequency is reduced, the AUC remains relatively stable.

*Figure 5.* Sensitivity to Neighbor Rebuild Interval.

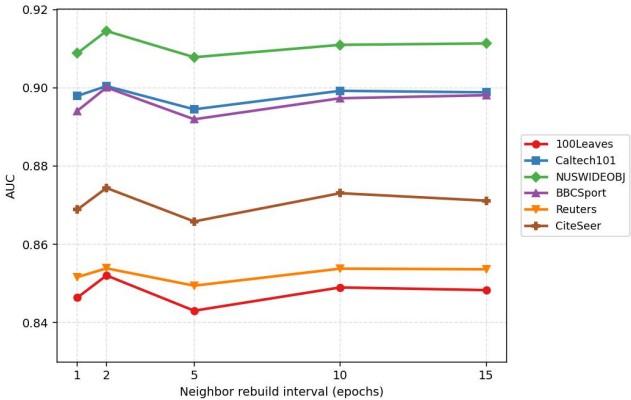

## B. Space Complexity

The memory is determined by storing the latent feature embeddings and the neighborhood indices.

1. **Latent Representations:** Storing the latent features $h_i^v \in \mathbb{R}^{D_h}$ for all $N$ samples in $V$ views requires $\mathcal{O}(VND_h)$

space.

2. **Neighborhood Indices:** Storing the indices of the $K$ reliable neighbors $\mathcal{N}_i^v$ for all samples in all views requires $\mathcal{O}(VNK)$ space.

The overall space complexity is:

$$\mathcal{O}(VND_h + VNK) = \mathcal{O}(VN(D_h + K)) \tag{17}$$

The space complexity is linear with respect to the number of samples $N$, confirming the model's memory efficiency.

## C. Supplementary Ablation Study

We additionally conduct several ablation studies to demonstrate the contribution of each module in our method. The results are reported in Table 6, where **AB** denotes the angle-based alignment loss, **DB** denotes the distance-based alignment loss, **AS** denotes the angle-based score, **DS** denotes the distance-based score, and **FS** denotes the full score. Note that the original design did not include a distance-based loss; therefore, the last column in Table 6 represents our model. From (A), (B), and (E), we observe that the direction-based alignment loss achieves the best performance. From (C), (D), and (E), it is evident that an effective outlier score must incorporate both components.

