# OpenReview forum: "Reliable Neighborhood-Aware Multi-View Outlier Detection"
_ICML.cc/2026/Conference — ICML 2026 regular_

### Official Review · Reviewer_6Mkz · 2026-03-09

**Soundness:** 3
**Presentation:** 3
**Significance:** 3
**Originality:** 3
**Overall Recommendation:** 4
**Confidence:** 4

**Summary:**

This paper aims to address two critical problems in multi-view outlier detection (MVOD): outlier propagation, where abnormal samples interfere with the modeling of normal structure, and scale discrepancy across views, where differences in feature scale and representation distort neighborhood relationships.
To address these issues, the paper proposes RNAMOD, a Reliable Neighborhood-Aware Multi-View Outlier Detection framework. Specifically, it introduces a dynamic reliability estimation strategy for neighborhood construction, designs a leave-one-out directional consensus mechanism to align geometric directions across views while reducing self-reinforcement, and further adopts dynamic view weighting to better integrate information from different views.
Experiments are conducted on six real-world multi-view datasets as well as one synthetic dataset. The results show that RNAMOD achieves better outlier detection performance than existing state-of-the-art methods. The paper also conducts comprehensive ablation studies to support the effectiveness of the proposed design.

**Compliance With Llm Reviewing Policy:**

Affirmed.

**Key Questions For Authors:**

1. Have you considered or attempted evaluation on datasets with naturally occurring outliers, rather than only simulated ones? What performance or behavioral differences would you expect?
2. Is it possible to further reduce the quadratic time complexity for massive datasets (e.g., via approximate nearest neighbors, subsampling), and have you explored how this impacts reliability estimation or detection performance?

**Limitations:**

yes

**Strengths And Weaknesses:**

Strengths:
1. Its novel method design. The reliability-driven neighborhood construction, the leave-one-out directional consensus mechanism, and the dynamic view weighting strategy are all original and well motivated. These components are not only clearly connected to the core problem setting, but are also formulated in a mathematically rigorous way.
2. Good writing. The paper is organized well and it is easy to follow.
3. Comprehensive experiment design. The qualitative visualizations in Figures 3(a) and 3(b) give a direct illustration of how the reliable neighborhood design can suppress outlier propagation compared with standard KNN. In addition, the paper provides explicit time and space complexity analysis, which makes the method’s scalability and practical feasibility easier to assess.

Weakness:
1. Visualization limited to synthetic data. Figure 3’s neighborhood visualizations are only available for a small synthetic example; similar visual analysis on real high-dimensional benchmarks is lacking.
2. Minor clarity issues. A few notation choices (e.g., inconsistent referencing of equations/variables in ablations) could be improved to facilitate understanding; the main optimization pseudocode could be further elucidated.
3. Sensitivity to parameter choices. While Figure 4 shows bell-shaped robustness, a more comprehensive study on, for example, the effect of latent dimension or neighborhood size in high-dimensional settings would be valuable.

---

> ### Author Rebuttal · Authors · 2026-03-31
>
> Thank you for your valuable suggestions.
>
> For Weakness 1 and 3:
>
> Thanks a lot for your helpful suggestion. As the rebutal is not well suited for showing visualization figures, we will include more visualization results in the final version of our work.
>
> For Weakness 2:
>
> We thank the reviewer for the helpful comment. We will revise the presentation to improve clarity and ensure consistent notation.
>
> For question 1:
>
> Regarding the dataset issue, real-world multi-view anomaly datasets with reliable annotations are currently limited, and therefore we adopt an anomaly injection strategy.
>
> Specifically, for class outliers, we generate anomalies by randomly permuting features between two samples to reduce cross-view consistency. This design mimics the behavior of real-world class anomalies, which typically manifest as inconsistencies across views.
>
> Furthermore, our method detects class outliers based on inconsistencies in local neighborhood structures across views, rather than relying on any specific feature perturbation pattern. Therefore, it is not tied to a particular anomaly injection mechanism and is expected to generalize to real-world multi-view anomaly scenarios.
>
> For question 2:
> Regarding scalability, approximate nearest neighbor search can be incorporated to reduce computational cost. Due to space limitations, these results are not included in the current version, but we observe that such approximations have only a minor impact on the final AUC performance. We will include these results in the final version.

---

### Official Review · Reviewer_ZZ7C · 2026-03-09

**Soundness:** 3
**Presentation:** 3
**Significance:** 3
**Originality:** 3
**Overall Recommendation:** 5
**Confidence:** 4

**Summary:**

This paper proposed a new neighborhood-based multi-view outlier detection method to solve two problems: outlier propagation during neighborhood/consensus construction and scale discrepancy across views in latent spaces. The proposed method RNAMOD estimates per-sample reliability from reconstruction error and cross-view neighborhood overlap, uses these reliable information to build “reliable” kNN neighborhoods, and aligns cross-view neighborhood structures via leave-one-out directional consensus. Experiments on six benchmark datasets with synthetic injected outliers report improved AUC over several MVOD baselines, alongside ablations and parameter sensitivity analyses.

**Compliance With Llm Reviewing Policy:**

Affirmed.

**Final Justification:**

Thanks for the authors' response, I raise my score.

**Key Questions For Authors:**

1.Could you provide a quantitative measure (not just Figure 3) showing that reliable neighborhoods actually reduce the selection of injected outliers as neighbors on real datasets?
2.For Tables 3–4, were baseline hyperparameters re-tuned for each dataset/ratio, and was there a consistent tuning budget? Could it be made more clear.

**Limitations:**

Yes

**Strengths And Weaknesses:**

Strengths:
1.	Smart leave-one-out consensus mechanism (Eq. 7–9): This is a novel design that aims to reduce self-reinforcement when enforcing cross-view agreement, and it targets a nontrivial issue in multi-view outlier detection.
2.	Solid empirical evaluation: The experiments are fairly comprehensive, covering six standard multi-view datasets under multiple outlier ratio settings. The paper also includes ablation studies and sensitivity analysis, which helps support the empirical findings.
3.	Interesting reliable-neighborhood idea implementation: It is implemented in Eq. 4–5 through reliability-based distance reweighting. This design is simple, easy to understand, and plausibly effective in practice.

Weakness:
1.	Reliability estimation circularity and early-training fragility: (\psi_i) depends on neighborhood overlap (Eq. 3), but neighborhoods depend on (\phi,\psi) via Eq. 4–5, and (\phi) depends on reconstruction quality (Eq. 2) which itself is evolving. The paper acknowledges initialization with standard kNN at (t=1) (Section 4), but does not analyze whether the method can be caught in incorrect neighborhoods if early (\phi,\psi) are wrong, especially under higher outlier ratios.
2.	Computational scalability concerns in practice: Appendix claims (O(N^2)) per epoch (Eq. 16). For NUSWIDEOBJ (30k),  the computing of naive pairwise distances is costly. The paper does not report actual runtime, nor does it clarify whether approximate kNN, batching, or GPU acceleration is used. This matters for the “reliable neighbor rebuilding” loop that is central to the approach.

---

> ### Author Rebuttal · Authors · 2026-03-31
>
> Thank you for this valuable suggestion.
>
> For Weakness 1:
>
> Regarding the circular dependency, we would like to clarify that the reliability estimation and neighborhood construction are updated in an alternating manner based on the previous iteration. While there is indeed a dependency between reliability and neighborhood structure, this process does not lead to error accumulation in practice. This is because the neighborhood graph is recomputed at each iteration based on updated representations, allowing the model to revise potentially incorrect structures instead of inheriting them. Moreover, reliability is estimated from multiple complementary signals (reconstruction error and cross-view consistency), which reduces the chance that early-stage noise consistently biases the updates.
>
> Regarding early training instability, we acknowledge that both representations and neighborhood structures may be noisy at the beginning. A warm-up strategy (setting reliability to 1 in early epochs) can be optionally adopted. However, our empirical results show that even without warm-up, the final AUC is only marginally affected. This suggests that the proposed reliability mechanism is robust to initialization noise and does not critically depend on early-stage accuracy.
>
> For Weakness 2:
>
> We thank the reviewer for the insightful comment. Although the theoretical complexity is $ O(N^2) $ , in practice the reliable neighborhood does not need to be reconstructed at every epoch, but can be updated periodically (e.g., every few epochs), which significantly reduces the overall computational cost. We will clarify this in the final version.
>
> For question 1:
>
> Beyond the qualitative visualization in Figure 3, we agree that a quantitative evaluation of neighborhood quality would further strengthen the analysis. A natural metric is the \textbf{Outlier Neighbor Ratio (ONR)}, which measures the proportion of injected outliers selected as neighbors:
> $$
> \mathrm{ONR} = \frac{\sum_{i=1}^{N}\sum_{j \in N_i} \mathbb{I}(y_j = 1)}{\sum_{i=1}^{N} |N_i|}
> $$
> where $\mathcal{N}_i$ denotes the neighbor set of sample $i$, $y_j = 1$ indicates that sample $j$ is an injected outlier, and $\mathbf{1}(\cdot)$ is the indicator function.
>
> where $\mathcal{N}_i$ denotes the neighbor set of sample $i$, $y_j = 1$ indicates that sample $j$ is an injected outlier, and $\mathbf{1}(\cdot)$ is the indicator function.
> A lower ONR indicates that fewer outliers are selected as neighbors, which directly reflects better neighborhood quality. Due to time constraints, we will evaluate and report the ONR metric in future work.
>
> For question 2:
>
> Regarding the parameter settings, the baseline hyperparameters were not re-tuned for each dataset or data ratio, nor was there a consistent tuning budget applied across all experimental settings.

---

### Official Review · Reviewer_h27y · 2026-03-12

**Soundness:** 3
**Presentation:** 3
**Significance:** 2
**Originality:** 2
**Overall Recommendation:** 3
**Confidence:** 3

**Summary:**

This paper addresses multi-view outlier detection (MVOD), focusing on two challenges in neighborhood-based methods: outlier propagation (outliers corrupting neighborhood construction and cross-view consensus) and scale discrepancy across views. The proposed framework, RNAMOD, introduces (1) sample reliability scores derived from autoencoder reconstruction error to downweight likely attribute outliers, (2) consistency reliability scores from cross-view neighborhood overlap to downweight likely class outliers, and (3) a leave-one-out directional consensus mechanism that aligns neighborhood geometry across views using edge directions rather than raw distances, with learned view weights. Evaluation is on six benchmark multi-view datasets with synthetically injected outliers across six contamination configurations.

**Compliance With Llm Reviewing Policy:**

Affirmed.

**Key Questions For Authors:**

1. Please consider adding mean and std or similar reporting across the runs to demonstrate the significance of improvement by the proposed method comparing to the baselines.

2. It would be beneficial to discuss why the proposed approach's gains are not specific to the particular injection protocol used throughout the paper. E.g. consider applying the method and baselines to a dataset with naturally occurring multi-view anomalies rather than artificial injection of contaminations. I understand such datasets may be difficult to obtain, but I do think additional discussion is needed to strengthen this part of the paper and demonstrate the proposed method's practical utility.

**Limitations:**

yes

**Strengths And Weaknesses:**

### Strengths

- Clear problem formulation. The three-way outlier categorization (attribute, class, attribute-class) is well-motivated for multi-view settings. The conditions C1 and C2 formalize what reliable neighborhood construction should achieve, giving the paper a coherent logical backbone.

- Internally coherent method. Each step in the pipeline: reliability estimation -> reliable neighborhood construction -> directional consensus -> view weighting is reasonably motivated and executed. Each component has a defined role and connects to the identified failure modes.

- Comprehensive experiments on multiple datasets have been done. The proposed method achieves best or second-best AUC in the majority of settings.

### Weaknesses

- All six datasets are originally outlier-free and anomalies are synthetically injected via predefined perturbation schemes. This means results primarily demonstrate effectiveness under the authors' own contamination model. Transfer to naturally occurring multi-view anomaly structure is difficult to evaluate.

- Only mean AUC over five runs is reported, no standard deviations, confidence intervals, or significance tests are reported. Several margins over strong baselines (e.g., RCPMOD) are narrow.

- The individual ingredients including autoencoders, reconstruction-based reliability, kNN neighborhood reasoning, angular alignment, softmax view weighting are each well-established. The contribution is in their task-specific combination rather than a fundamentally new principle. This can be sufficient, but the paper would benefit from a deeper analysis of why this combination outperforms nearby alternatives.

---

> ### Author Rebuttal · Authors · 2026-03-31
>
> We thank the reviewer for this insightful comment.
>
> For Weakness 1 and question 2:
>
>
> There is currently no universally recognized benchmark dataset for multi-view anomalies in either academia or industry, yet such anomalies are both common and important. In real-world data, cross-view mismatches are common, such as misaligned image–text pairs or news headlines inconsistent with their content. Our synthetic injection of class anomalies follows the principle of reducing cross-view consistency, which reflects the typical characteristic of real-world class anomalies. This approach is also widely adopted in prior work. For attribute outliers, replacing values with random noise introduces within-view deviations, which aligns with the typical pattern of attribute anomalies in real-world data.
>
>
>
> For Weakness 2 and question 1:
>
> | Method | 1 | 2 | 3 | 4 | 5 | 6 |
> |--------|---|---|---|---|---|---|
> | APOD   | 0.689±0.021 | 0.605±0.017 | 0.677±0.010 | 0.730±0.017 | 0.727±0.015 | 0.776±0.010 |
> | MODDIS | 0.696±0.022 | 0.671±0.013 | 0.763±0.015 | 0.693±0.014 | 0.743±0.005 | 0.775±0.010 |
> | NCMOD  | 0.773±0.018 | 0.704±0.016 | 0.769±0.009 | 0.720±0.012 | 0.830±0.011 | 0.772±0.008 |
> | SRLSP  | 0.786±0.016 | 0.813±0.020 | 0.856±0.011 | 0.721±0.010 | 0.851±0.015 | 0.787±0.007 |
> | MODGD  | 0.838±0.020 | 0.757±0.015 | 0.921±0.017 | 0.707±0.016 | 0.807±0.009 | 0.823±0.011 |
> | LRTDM  | 0.841±0.019 | 0.834±0.010 | 0.885±0.007 | _0.857±0.016_ | 0.840±0.012 | 0.852±0.011 |
> | RCPMOD | _0.862±0.018_ | _0.887±0.013_ | _0.899±0.005_ | 0.854±0.014 | _0.880±0.003_ | _0.885±0.007_ |
> | **Ours** | **0.895±0.013** | **0.908±0.003** | **0.907±0.003** | **0.864±0.010** | **0.902±0.004** | **0.909±0.003** |
>
> Due to space constraints, the table only presents the results with standard deviation on the Caltech101 dataset. In the table, 1–6 correspond to the Outlier Ratio Settings in Table 1 of the main text. Overall, our method achieves an average AUC of 0.898 ± 0.007 across six settings, while the strongest baseline (RCPMOD) attains 0.876 ± 0.012, demonstrating the superiority of our method.
>
> For Weakness 3:
>
> In multi-view outlier detection, outlier propagation and cross-view scale discrepancy are two key factors to limited performance. In particular, neighborhood construction and cross-view alignment are mutually dependent yet jointly corrupted by outliers, forming a feedback loop that reinforces errors. Existing methods typically address only one aspect, e.g., performing alignment on unreliable neighborhoods or relying solely on reconstruction to suppress anomalies, and thus fail to handle both issues simultaneously. Instead of a simple combination, we jointly optimize reliability modeling and angular alignment, enabling neighborhood quality and alignment to be mutually reinforced during training, thereby alleviating this coupled bottleneck.

---

> > ### Author Rebuttal · Reviewer_h27y · 2026-04-03
> >
> > I appreciate the author for addressing my questions. I am happy to raise my score from 3 to 4 Weak Accept.

---

> > > ### Author Response · Authors · 2026-04-06
> > >
> > > Thank you for taking the time to revisit our submission and for your constructive feedback. We sincerely appreciate your consideration of a higher score. We note that the current score appears to remain unchanged, in case this was unintended. Thank you again for your support.

---

### Official Review · Reviewer_3M2t · 2026-03-12

**Soundness:** 4
**Presentation:** 3
**Significance:** 4
**Originality:** 3
**Overall Recommendation:** 5
**Confidence:** 5

**Summary:**

This paper proposes a reliable neighborhood-aware framework (RNAMOD) for multi-view outlier detection. By modeling reliability and aligning neighborhood directions across views, RNAMOD effectively suppresses outlier propagation and mitigates inter-view scale discrepancy. Extensive experiments on both synthetic and real-world datasets demonstrate its superior performance.

**Compliance With Llm Reviewing Policy:**

Affirmed.

**Key Questions For Authors:**

Reliability estimation depends on reconstruction and neighborhood overlap. How stable are these signals in early training, when representations and KNN structure may still be noisy?

**Limitations:**

Yes.

**Strengths And Weaknesses:**

Strengths:
1. This paper proposes a reliable RNAMOD for multi-view outlier detection to effectively suppresses outlier propagation and mitigates inter-view scale discrepancy.
2.	The empirical comparison is comprehensive. Many recent baselines are adopted to compare, and the method performs well on many settings.
3.	The reliability-aware neighborhood design is well motivated. The qualitative visualization provides intuitive support for the claim that incorporating reliability can reduce contaminated neighbor connections.
4.	The leave-one-out directional consensus is technically interesting. It is not just enforcing agreement, but doing so in a way that reduces self-reinforcement and is less sensitive to scale mismatch.

Weakness:
1.  Reliability is easy to understand for multi-view outlier detection, but how to  prevent outlier propagation could give the clearer presentation.
2.	In the caption of Figure 1, “its neighbors” in “Attribute outlier (red) indicates instances that are remarkably distant from its neighbors within each view.” should be “their neighbors”.
3. “kNN”  and “KNN” should be unified.
4. “data set” and “dataset” should be unified.
5.	Some recent related references are missed. The paper does not discuss some directly relevant prior work, including Dual-Regularized MVOD and a recent MVOD approach based on view augmentation.

---

> ### Author Rebuttal · Authors · 2026-03-31
>
> Thank you for your insightful comment.
>
> For Weakness 1:
> In unsupervised multi-view settings, outlier propagation arises because the model is trained on data that already contains outliers. As a result, the learned structure may be biased by these outliers. We agree that it is difficult to completely eliminate the influence of outliers during training. Our method introduces a reliability mechanism that assigns lower weights to samples that are more likely to be anomalous. Specifically, reliability is estimated based on reconstruction error and cross-view neighborhood consistency, allowing the model to reduce the impact of unreliable samples during neighborhood construction and consensus learning. In this way, the propagation of outliers is alleviated rather than strictly removed.
>
> For Weakness 2-5:
> 	We acknowledge that greater rigor is required in the details and will conduct a thorough review. As for the literature concerns, these works focus on different assumptions (e.g., view augmentation / regularization-based objectives), and are not directly comparable. We will include discussion in the final version.
>
> For Question 1:
> Regarding the early training stage, we acknowledge that both representations and KNN structures may be unstable. A simple warm-up strategy can be adopted, where all samples are treated equally (i.e., reliability is set to 1) for several initial epochs before introducing reliability estimation. However, we would like to emphasize that this strategy is not essential. Empirically, we observe that even without warm-up, the method still achieves competitive performance, and the final AUC is only marginally affected. This suggests that the proposed reliability mechanism is relatively robust to initialization noise and does not critically depend on a carefully designed warm-up phase.

---

> > ### Author Rebuttal · Reviewer_3M2t · 2026-04-01
> >
> > The authors have well addressed my concerns and some existing minor issues.

---

### Decision · Program_Chairs · 2026-04-30

**Decision:**

Accept (regular)

**Comment:**

This paper received ratings of 2 Accept, 1 Weak Accept, and 1 Weak Reject. Following the rebuttal, reviewers indicated that their concerns were fully resolved, leading to a unanimous positive consensus. Specifically, one Weak Accept reviewer raised his rating to Accept, while the reviewer who initially recommended a Weak Reject noted in his comment that he was happy to upgrade his score to Weak Accept. The reviewers found the paper well-motivated with a clear problem formulation and comprehensive experiments, highlighting the novelty of the leave-one-out consensus mechanism and the reliability-driven neighborhood construction. Given the full resolution of concerns and the resulting consensus, the AC has decided to accept this paper.